# Adsorption Properties of Pd_3_-Modified Double-Vacancy Defect Graphene toward SF_6_ Decomposition Products

**DOI:** 10.3390/s20154188

**Published:** 2020-07-28

**Authors:** Jie Li, Lei Pang, Fuwei Cai, Xieyu Yuan, Fanyu Kong

**Affiliations:** School of Electrical Engineering, Xi’an Jiaotong University, Xi’an 710049, China; jl1562074348@stu.xjtu.edu.cn (J.L.); za41255@stu.xjtu.edu.cn (F.C.); yxy201115010@163.com (X.Y.); kfy3119104175@stu.xjtu.edu.cn (F.K.)

**Keywords:** Pd_3_-graphene, SF_6_ decomposition products, DFT, adsorption

## Abstract

In this study, we investigate Pd_3_-cluster-modified 555–777 graphene (Pd_3_-graphene) as a novel resistor-type gas sensor to detect SF_6_ decomposition products based on density functional theory calculations. We obtained and minutely analyzed the relevant parameters of each most stable adsorption configuration to explore the microscopic mechanism during gas adsorption. Theoretical results reveal that Pd_3_-graphene shows great adsorption capacity and sensitivity toward those decompositions. High adsorption energies and abundant charge transfer amounts could guarantee a stable adsorption structure of decomposition gases on Pd_3_-graphene surface. The complex change of density of states verifies a strong chemical reaction between the gases and the surface. Moreover, the conductivity of Pd_3_-graphene would improve due to the decrease of energy gap, and the sensitivity was calculated as SOF_2_ > H_2_S > SO_2_ > SO_2_ F_2_. This work provides an effective method to evaluate the operation status of SF_6_ gas-insulated equipment.

## 1. Introduction

Owing to its outstanding insulation and arc-extinguishing properties, SF_6_ has been widely applied in gas-insulated switchgear (GIS) [1,2]. However, in the long-term operation of GIS, several inevitable insulation defects may cause partial discharge and partial overheat, which will lead to the initial decomposition of SF_6_ [3,4]. Simultaneously, SF_6_ will ultimately decompose to several characteristic products with the reaction of trace O_2_ and H_2_O in SF_6_-insulated equipment: H_2_S, SO_2_, SOF_2_ and SO_2_ F_2_ gas [5,6,7]. These decomposition products can corrode the metal parts inside the equipment, accelerate the aging of the insulation medium—and even result in the sudden failure of GIS [8,9]. Therefore, discovering SF_6_ decomposition products and dealing with insulation defects in a timely manner have great importance. To date, several methods such as gas chromatography, mass spectrometry and Fourier transform infrared spectroscopy, are suggested to detect SF_6_ products [10,11]. However, these methods are either inaccurate or complicated, so none of them are used in detecting SF_6_ decomposition products.

In recent years, the gas sensor method has been used in various fields due to its advantages such as high sensitivity, rapid response and small size. Graphene is a 2D material with a unique 2D monoatomic layer structure and an electronic energy band structure. Its excellent characteristics—including high electron mobility, high thermal conductivity, brilliant mechanical properties and large specific surface area—make it a promising gas sensor material [12,13,14,15,16,17]. However, in actual production, several intrinsic defects in graphene may be observed, such as single defects, double-vacancy defects and Stone–Wales defects [18]. Lattice defects may cause local charge traps on the graphene surface, which has a substantial effect on improving the electronic structure and the adsorption capacity of graphene. Liu et al. anchored Sc, Zn, Mo, Ru, Rh, Pd and Ag atoms on defective graphene for N_2_ reduction reaction and found that a single Mo embedded on nitrogen-doped 555–777 graphene shows eminent catalytic performance [19]. Arokiyanathan et al. calculated the adsorption properties of small Li clusters on graphene with Stone–Wales defect. The results showed that Li clusters have a strong interaction with the defect region [20]. Ma et al. reported hydrogen adsorption on Co-4-doped defective graphene and discovered that point defects in graphene can effectively improve the hydrogen storage capacity of Co-4 [21]. Despite much research on defective graphene [22,23,24,25], few relevant research on SF_6_ decomposition product detection have been reported.

555–777 graphene is a double-vacancy defect graphene sheet [18]. The absence of two carbon atoms does not destroy the original SP^2^ hybrid orbital network but forms a stable topological hole. Thus, 555–777 graphene has a more stable electronic structure compared with other defective structures. Previous studies showed that doping transition metal elements on graphene surface can improve its electrical conductivity and enhance its capacity to adsorb gas molecules [26,27,28,29,30]. Liu et al. calculated the adsorption of SF_6_ decompositions on Pd (111) surface and found that Pd (111) would have great adsorption capacity on absorbing those products. Thus, doping Pd element on 555–777 graphene may be a potential method to improve its adsorption properties [30]. In this study, Pd_3_-doped 555–777 graphene (Pd_3_-graphene) is carried out to detect SF_6_ decomposition products (H_2_S, SO_2_, SOF_2_ and SO_2_ F_2_). Different adsorption structures of each gas are obtained to find the most stable adsorption structure. Furthermore, adsorption energy, charge transfer, density of states (DOS) and deformation charge density (DCD) of Pd_3_-graphene surface before and after gas adsorption are calculated to reveal the gas response mechanism. Overall, our work aims to provide some theoretical basis for developing a novel graphene sensor.

## 2. Computational Details and Models

In this work, all calculations are carried out using Dmol^3^ package based on density functional theory (DFT) [31]. Generalized gradient approximation and Perdew–Burke–Ernzerhof function are utilized to deal with the electron exchange-correlation energy and interaction effect of electrons [32,33]. DFT semi-core pseudopots are set to recoup the relativistic effect of Pd atoms. The Brillouin zone is sampled with 6 × 6 × 1 k-points in the Monkhorst–Pack grid [34]. Double numeric plus polarization basis sets are used to obtain a higher calculation precision of hydrogen bond [35]. The smearing is set as 0.001 Ha to ensure the accuracy of the DFT calculation. The SCF tolerance is 1 × 10^−6^ Ha and spin polarization is considered due to the magnetic properties of Pd atoms. Additionally, total energy of convergence tolerance, maximum force and maximum displacement are set as 1 × 10^−5^ Ha, 2 × 10^−3^ Ha/Å and 5 × 10^−3^ Å, respectively.

Intrinsic 555–777 graphene surface consists of a 6 × 6 supercell, and the lattice parameter is 14.760 Å × 14.760 Å. The vacuum area is set as 20 Å to avoid the interaction of neighboring slabs. The Pd_3_ cluster is a triangular structure composed of three Pd atoms. Through DFT calculations, various parameters of stable adsorption structures can be acquired, including adsorption energy (*E*_ad_), charge transfer amount (*Q*_T_), adsorption distance (D) and energy gap (*E*_g_). The definition of these parameters are the same as the previous study [36].

## 3. Results and Discussions

### 3.1. Geometric Optimization

We apply geometric optimization to minimize their energy and obtain the most stable status of H_2_S, SO_2_, SOF_2_, SO_2_ F_2_ and Pd_3_-graphene. The optimized structures are shown in Figure 1 (adapted from [36]). Figure 1a shows the stable structure of intrinsic 555–777 graphene with three pentagons and heptagons at the center of the graphene sheet. Considering that the charge trap in the center of the hole has a strong binding ability to electrons and the special D electronic structure of Pd metal, we dope the Pd_3_ cluster at the defect center of graphene to form a stable anchored structure. Figure 1b shows that the Pd_3_ cluster is parallel to the graphene surface, indicating that the force of the crystal plane on the three Pd atoms is almost equal. Bond lengths between the Pd atoms are 2.714, 2.714 and 2.812 Å. Millikan charge analysis shows that 0.16 electrons transfer from the Pd_3_ cluster to the graphene sheet after geometric optimization, showing an evident orbital hybridization between C and Pd atoms. The electronic structure of the graphene surface will be greatly changed, which may substantially improve gas adsorption capacity.

### 3.2. Adsorption Systems

#### 3.2.1. H_2_S Adsorption on Pd_3_-Graphene

The H_2_S molecule is placed in different distances and angles to approach the Pd_3_-graphene surface and obtain the most stable adsorption structure of H_2_S gas. Considering the symmetry of H_2_S molecule, its two typical stable adsorption structures are acquired through H and S atoms approaching the Pd_3_-graphene surface, as shown in Figure 2. *E*_ad_, *Q*_T_ and D are listed in Table 1.

Figure 2a shows the adsorption configuration of system M1. The H_2_S molecule approaches the surface by the S atom and forms a stable interaction between the S and Pd atoms. The D of M1 is 2.338 Å, revealing that the reaction is relatively strong. When the adsorption energy is greater than 0.6 eV, the adsorption is generally considered chemical. The corresponding *E*_ad_ of M1 is −1.211 eV; hence, the adsorption is chemical, and the process is spontaneous. The positive value of *Q*_T_ of M1 (0.286) demonstrates that 0.286 electrons transfer from the H_2_S molecule to the Pd_3_-graphene.

In Figure 2b, the H_2_S molecule approaches the Pd_3_-graphene by H atom, forming a new ionic bond with the Pd atom. The length of the H–Pd bond is 1.868 Å, indicating a strong chemical interaction between H and Pd atoms. Similar to system M1, 0.279 electrons transfer from the H_2_S molecule to the sheet during optimization. The *E*_ad_ of system M2 (−1.185 eV) is smaller than that of M1 (−1.211 eV), which means that system M1 releases more energy than M2 after H_2_S adsorption. Above all, according to the greater *E*_ad_ and *Q*_T_ of configuration M1, M1 is the most stable adsorption structure of H_2_S on Pd_3_-graphene surface.

DOS and partial density of states (PDOS) of the most stable adsorption structure (M1) are calculated to analyze the mechanism of H_2_S adsorption further, as shown in Figure 3. In Figure 3a, the black and red lines represent the DOS before and after H_2_S adsorption, respectively. Several distinct changes are observed in DOS after H_2_S adsorption, which mainly reflect in the increase of DOS from −8 eV to −4 eV. The DOS at the Fermi level rises slightly, revealing that electrons can more easily transfer from the valence band to the conduction band, which may have a substantial effect on improving its conductivity. The PDOS in Figure 3b shows that the increased area of DOS from −8 eV to −4 eV mainly consists of the 3p orbit of S atom. The large overlapped area from −7 eV to −3 eV between the 4d orbit of Pd atom and the 3p orbit of S atom manifests that the chemical reaction between these atoms is strong, which is consistent with the above analysis.

#### 3.2.2. SO_2_ Adsorption on Pd_3_-Graphene

For the adsorption of SO_2_ on Pd_3_-graphene surface, various initial positions of SO_2_ molecule are set to investigate the most stable adsorption system. After geometric optimization, two typical stable adsorption configurations are obtained. Figure 4 shows the adsorption structures through S and O atoms approaching the sheet. Figure 4a shows that the SO_2_ molecule forms a stable adsorption structure by building a chemical bond between S and Pd atoms. The length of the S–Pd bond is 2.192 Å, declaring a strong interaction between them. The two O atoms are far from the sheet; thus, no direct interaction may exist between them and the crystal plane. In Figure 4b the SO_2_ molecule approaches the Pd_3_ cluster by O atoms, forming two chemical bonds with the Pd atoms. The length of both O–Pd bonds is 2.131 Å, which is shorter than that of the S–Pd bond in system M3. Thus, the chemical reaction in system M4 may be stronger than that in M3.

Table 2 presents the related parameters of configuration M3 and M4. Both their *E*_ad_ are negative, manifesting that the adsorption is exothermic and proceeds spontaneously. By contrast, the *E*_ad_ of system M4 is greater than that of system M3, implying that the SO_2_ molecule absorbed on the Pd_3_-graphene surface through two O atoms is the most stable structure. Given the strong reaction between O and Pd atoms in system M4, 0.283 electrons transfer from the sheet to SO_2_ molecule. For system M3, the small *Q*_T_ means that the ionic S–Pd bond is relatively easy to break.

The DOS and PDOS of M4 are intensively calculated to analyze the adsorption mechanism because M4 is the most stable adsorption configuration of SO_2_ on Pd_3_-graphene. Figure 5a presents that the DOS of M4 exhibits three new peaks after SO_2_ adsorption at around −8, −6 and −4 eV. PDOS shows that the new peaks mainly result from the 2p orbit of O atom and the 3p orbit of S atom. The small increase of DOS at fermi level demonstrates an increase of the conductivity of Pd_3_-graphene. In addition, a large overlapped area between the 2p orbit of O atom and the 4d orbit of Pd atom ranges from −5 eV to −3 eV, reflecting a complex hybridization of those orbits. Therefore, the chemical reaction between O atoms and Pd atom is strong enough to form a stable adsorption configuration.

#### 3.2.3. SOF_2_ Adsorption on Pd_3_-Graphene

Subsequently, we place the SOF_2_ molecule in different directions and distances approaching the sheet to explore the most stable adsorption structure. Figure 6 shows three typical adsorption configurations, in which the SOF_2_ molecule is absorbed on Pd_3_-graphene surface by S, F and O atoms. Related parameters are listed in Table 3.

Figure 6a shows that the SOF_2_ molecule approaches the surface by the S atom and forms a stable adsorption structure ultimately. The S atom is trapped by the Pd atom and builds a new chemical bond with a bond length of 2.170 Å. The *E*_ad_ of system M5 listed in Table 3 is −1.230 eV, which indicates that the adsorption is chemical, and the structure is stable enough. During adsorption, only 0.038 electrons transfer from SOF_2_ molecule to the sheet, so their electronic structure may change slightly. For system M6 shown in Figure 6b, SOF_2_ molecule absorbs on the Pd_3_-graphene surface by forming an ionic bond length of 2.124 Å between F and Pd atoms. The *E*_ad_ of M6 (−0.284 eV) is smaller than that of M5 (−1.230 eV), revealing that M5 is more stable than M6. In system M7, the SOF_2_ molecule absorbs directly above the hollow position of Pd_3_ cluster. The length of the O–Pd bond is 2.299 Å, which is longer than that of M5 and M6. However, the specific *E*_ad_ of M7 (−1.282 eV) is greater than that of M5 (−1.230 eV), implying that system M7 is more stable. Opposite to system M5, 0.132 electrons transfer from the sheet to SOF_2_ molecule after absorbing on Pd_3_-graphene. Hence, the SOF_2_ molecule tends to be absorbed on the Pd_3_-graphene surface in the form of M7, establishing an adsorption system with the lowest energy.

Figure 7 shows the DOS and PDOS of the most stable adsorption structure of SOF_2_ on the Pd_3_-graphene surface. The DOS of M7 structure increases distinctly in the range from −9 eV to −7.5 eV and −6 eV to −3 eV after SOF_2_ adsorption. These changes are mainly caused by the 2p orbit of O atom and the 2p orbit of F atom. A slight increase of DOS near the fermi level is observed, signifying that SOF_2_ adsorption can improve the conductivity of Pd_3_-graphene. The PDOS displaces the hybridization of individual atoms during adsorption. The 2p orbit of O atom, the 2p of F atom and the 4d orbit of Pd atom overlap from −6 eV to −1 eV, suggesting that the chemical reaction between them is considered strong, and the electronic structure of the SOF_2_ molecule is active.

#### 3.2.4. SO_2_ F_2_ Adsorption on Pd_3_-Graphene

Eventually, the SO_2_ F_2_ molecule approaches the Pd_3_-graphene surface in various directions and distances to find the most stable adsorption structure. After adsorption structure optimization, two typical adsorption systems are obtained, as shown in Figure 8. Correlative parameters of adsorption systems are listed in Table 4.

Figure 8a presents the adsorption configuration of SO_2_ F_2_ molecule approaching the Pd_3_-graphene surface by F atom. The Pd atom builds a chemical connection with the F atom during the approaching process and forms an F–Pd bond with length of 2.014 Å ultimately. The S–F bond is stretched to 2.40 Å due to the radial force of the Pd atom. In system M9, the SO_2_ F_2_ molecule approaches the sheet by O atom. Similarly, the O atom forms a chemical bond with the Pd atom, but the length of the O–Pd bond (2.181 Å) is longer than that of the F–Pd bond (2.014 Å), implying a stronger interaction in configuration M8. The *E*_ad_ of M8 (−0.920 eV) is greater than that of M9 (−0.804 eV), demonstrating that the M8 structure is more stable than the M9 structure. Additionally, 0.705 electrons transfer from the sheet to the SO_2_ F_2_ molecule in M8, which is 2.4 times of M9, confirming a stronger oxidation of the F atom. Thus, configuration M8 is supposed to be the most stable adsorption structure of SO_2_ F_2_ on the Pd_3_-graphene surface.

Figure 9a shows that DOS increases substantially in the range of −7 to 0.5 eV. PDOS reveals that the 2p orbit of O atom and the 2p orbit of F atom mainly contribute the increased area, that is, the DOS near the fermi level rises slightly. Thus, the conductivity of Pd_3_-graphene is expected to improve after SO_2_ F_2_ adsorption. According to the PDOS, a large hybridization is observed between the 2p orbit of F atom and the 4d orbit of Pd atom from −5 eV to −1 eV, implying that these orbits are relatively active, while SO_2_ F_2_ approaches the surface.

#### 3.2.5. SF_6_ Adsorption on Pd_3_-Graphene

In order to ensure that the Pd_3_-graphene sensor can be utilized in GIS, we further calculated the adsorption of SF_6_ gas on Pd_3_-graphene surface to eliminate the interference of SF_6_ gas. Same as other gases, SF_6_ approaches the graphene surface at different angles and distances. After the geometric optimization, we obtained the most stable adsorption structure of SF_6_ on Pd_3_-graphene surface shown in Figure 10. As presented, SF_6_ molecule approaches the Pd_3_-graphene surface by F atom and forms a weak interaction with the Pd atom. The adsorption distance of SF_6_ is 4.060 Å, which is longer than other gases. Moreover, the *E*_ad_ of SF_6_ is calculated to be–0.124 eV, reflecting that the adsorption the adsorption system is not stable enough. During the adsorption, 0.162 electrons transfer from the Pd_3_-graphene to SF_6_ molecule. In conclusion, the Pd_3_-graphene can be a potential sensor applied in the SF_6_ gas-insulated equipment.

### 3.3. Electronic Properties

The deformation charge density of pure Pd_3_-graphene and the four most stable adsorption systems is calculated to investigate the difference in electronic structure before and after modifying or gas adsorption. As presented in Figure 11, the red region means an increase of charge density after adsorption, whereas the blue region means a decrease. In Figure 11a, the charge density of Pd_3_ cluster decreases, revealing that the defect of 555–777 graphene has a strong oxidation during the doping process. Although the absence of carbon atoms do not give rise to dangling bonds, the reconstruction of the graphene structure results in the change of bond lengths. Meanwhile, the defect would cause a rehybridization of the sigma and pi orbitals of carbon atoms, which would enhance the electron activity near the defect. When the transition metal approaches the defect, its special d electrons form a strong chemical interaction with the defect. Figure 11b shows that the charge density of the two Pd atoms decreases, while the S atom receives electrons from the Pd_3_ cluster after adsorption. Therefore, the adsorption reaction is concentrated between the S atom and the Pd atom. In Figure 11c, electrons transfer from the Pd atoms to the O atom due to the strong electronegativity of the O atom, indicating that the two O–Pd bonds are stable enough to support the adsorption structure. The O atom also receives electrons from the S atom, such that the electronic structure inside the SO_2_ gas changes prominently. As for SOF_2_ gas, the charge density neighboring the O atom increases to a certain degree, and the charge density of the Pd atom decreases. In principle, the O atom performs as an electron acceptor during SOF_2_ adsorption. In Figure 11e, the DCD of system M8 shows that the F atom receives electrons from the Pd atom and the nearby the S atom, verifying the fairly strong oxidability of the F atom. In conclusion, the Pd_3_ cluster behaves as an electron donator during gas adsorption and leads to a violent chemical reaction in the vicinity, certifying that the Pd_3_ dopant can substantially enhance the adsorption capacity of intrinsic 555–777 graphene.

According to the evident effect on the electronic structure of Pd_3_-graphene after SF_6_ product adsorption, the sensitivity and selectivity for application of chemical gas sensor must be further investigated. As a resistor-type gas sensor, the change in conductivity is an important factor in detecting SF_6_ decomposition products. The conductivity (σ) of Pd_3_-graphene gas sensor could be evaluated by the following formula [37]:$$\sigma \;\propto {\;e}^{\left(-\frac{E_{g}}{2kT}\right)},$$
where *k*, *T* and *E*_g_ represent the Boltzmann constant, temperature and HOMO–LUMO energy gap, respectively. Therefore, under a certain temperature condition, σ is an exponential function of the *E*_g_ and a smaller energy gap would determine a higher conductivity. We perform frontier molecular orbital theory to calculate the energy of HOMO (*E*_H_) and LUMO (*E*_L_) and obtain the specific *E*_g_, which the difference between them.

Figure 12 intuitively presents the *E*_H_, *E*_L_ and *E*_g_ of pristine Pd_3_-graphene and the four most stable adsorption systems, where the black number represents the *E*_g_. For pristine Pd_3_-graphene, the *E*_g_ is 0.086 eV, signifying that the basic conductivity is relatively high, which probably results from the Pd_3_ dopant. After gas adsorption, the *E*_g_ of different systems decreases without exception. While H_2_S and SO_2_ absorbing on the Pd_3_-graphene surface in the most stable configuration, the *E*_g_ decreases by 13.95% and 9.30%, respectively, manifesting an increase in conductivity of the Pd_3_-graphene sensor. For the SOF_2_ adsorption system, the *E*_g_ of Pd_3_-graphene decreases by 38.37%, which is the largest among the four adsorption systems, indicating that the electronic structure is more continuous, and the electronic transition is effortless. However, for the SO_2_ F_2_ adsorption system, *E*_g_ reduces with the smallest degree, only 5.81%, which means the conductivity of the Pd_3_-graphene sensor has a slight rise. In conclusion, the conductivity of Pd_3_-graphene sensor increases after gas adsorption, which agrees with the analysis in the DOS part.

As for N_3_&Ni doped 555–777 graphene reported in [36], N_3_&Ni-graphene has a high sensitivity on absorbing H_2_S and SO_2_ gas while the *E*_g_ decreases from 0.426 eV to 0.052 eV and 0.138 eV, respectively. Hence, the sensitivity of N_3_&Ni-graphene may be higher than Pd_3_-graphene while absorbing H_2_S and SO_2_ gas. However, it is toilsome for N_3_&Ni-graphene to distinguish SOF_2_ molecule and SO_2_ F_2_ due to their similar *E*_g_. The problem can be solved by using Pd_3_-graphene gas sensor for its obviously different *E*_g_ of various adsorption systems. As a result, Pd_3_-graphene can be a better gas sensor material on detecting SF_6_ decomposition product.

## 4. Conclusions

In this work, we propose Pd_3_-cluster-doped 555–777 graphene as a novel resistor-type gas sensor to detect SF_6_ decomposition products (H_2_S, SO_2_, SOF_2_ and SO_2_ F_2_), and the calculation results are as follows:

(I) Adsorption energies of the most stable adsorption systems of H_2_S, SO_2_, SOF_2_ and SO_2_ F_2_ are −1.211, −1.591, −1.282 and −0.920 eV, respectively, demonstrating that the four gases could be absorbed on Pd_3_-graphene surface by chemisorption. Charge transfer makes various atoms form ionic bonds during adsorption. In addition, the change of DOS and PDOS verifies high hybridizations of different atomic orbits.

(II) The conductivity of Pd_3_-graphene would be improved without exception due to the decrease of *E*_g_ after gas adsorption. SOF_2_ gas adsorption leads to the largest increase, whereas SO_2_ F_2_ is the smallest. According to the reduced value of *E*_g_, the sensitivity of the four gases follows the order SOF_2_ > H_2_S > SO_2_ > SO_2_ F_2_.

Thus, we confirm that Pd_3_-graphene can absorb SF_6_ decomposition products, and the Pd_3_-graphene sensor can be employed to evaluate the insulation condition of GIS by detecting these decompositions.

## Figures and Tables

**Figure 1 sensors-20-04188-f001:**
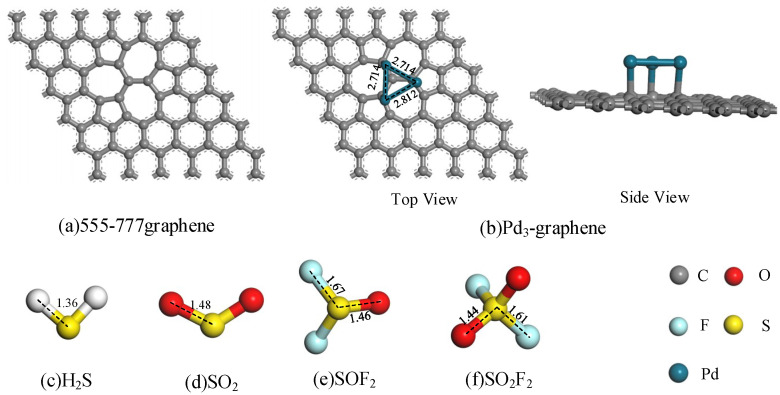
Optimized structures of intrinsic 555–777 graphene, Pd_3_-graphene and gas molecules (distance in Å). © 2020 IEEE.

**Figure 2 sensors-20-04188-f002:**
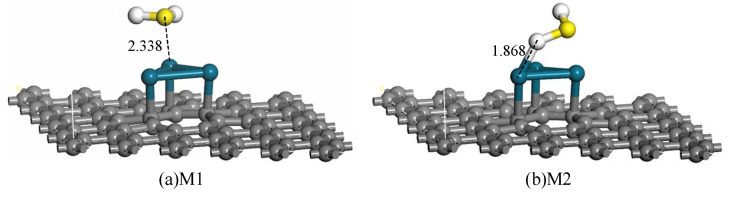
Adsorption structures of H_2_S on Pd_3_-graphene surface (distance in Å).

**Figure 3 sensors-20-04188-f003:**
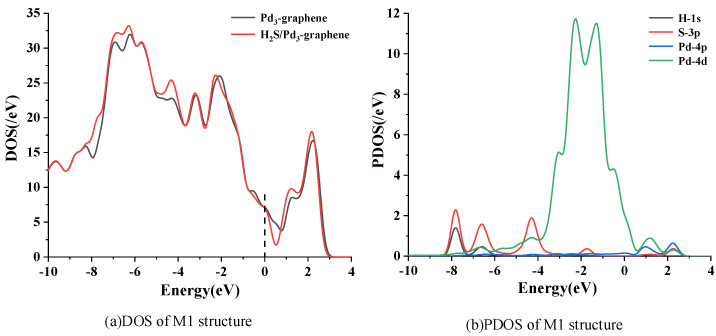
Density of states (DOS) and partial density of states (PDOS) of system M1. Dashed line represents the Fermi level.

**Figure 4 sensors-20-04188-f004:**
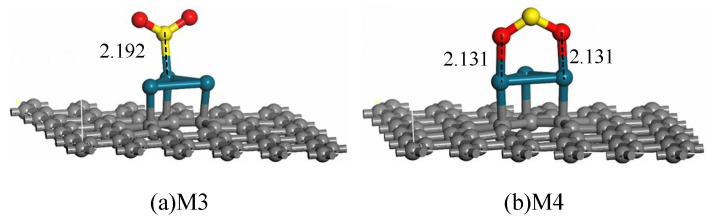
Adsorption structures of SO_2_ on Pd_3_-graphene surface (distance in Å).

**Figure 5 sensors-20-04188-f005:**
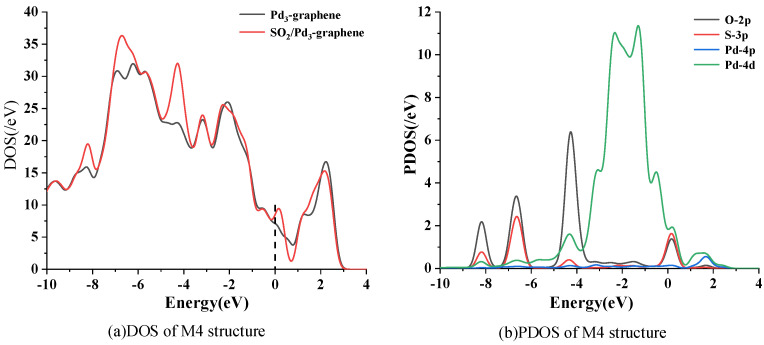
Density of states (DOS) and partial density of states (PDOS) of system M4. Dashed line represents the Fermi level.

**Figure 6 sensors-20-04188-f006:**
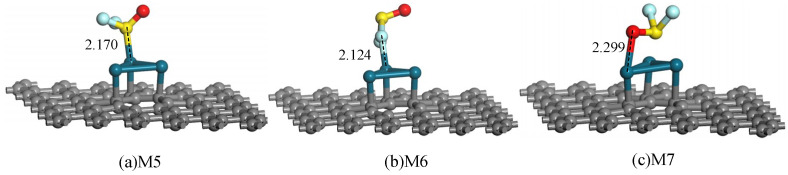
Adsorption structures of SOF_2_ on Pd_3_-graphene surface (distance in Å).

**Figure 7 sensors-20-04188-f007:**
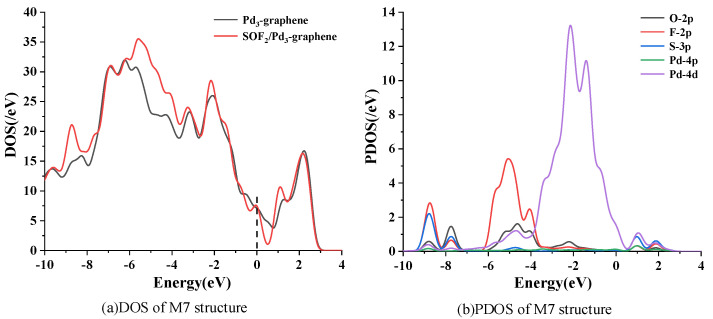
Density of states (DOS) and partial density of states (PDOS) of system M7. Dashed line represents the Fermi level.

**Figure 8 sensors-20-04188-f008:**
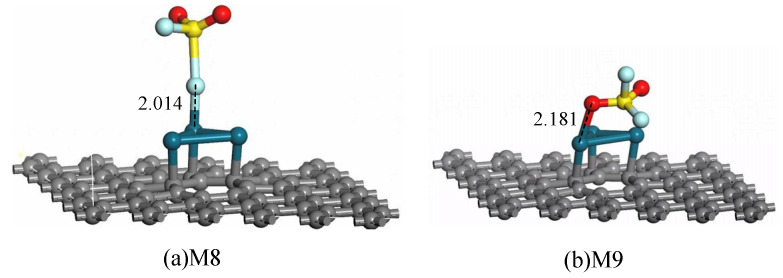
Adsorption structures of SO_2_ F_2_ on Pd_3_-graphene surface (distance in Å).

**Figure 9 sensors-20-04188-f009:**
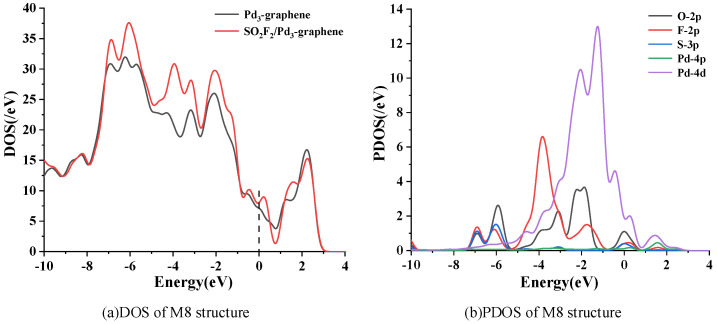
Density of states (DOS) and partial density of states (PDOS) of system M8. Dashed line represents the Fermi level.

**Figure 10 sensors-20-04188-f010:**
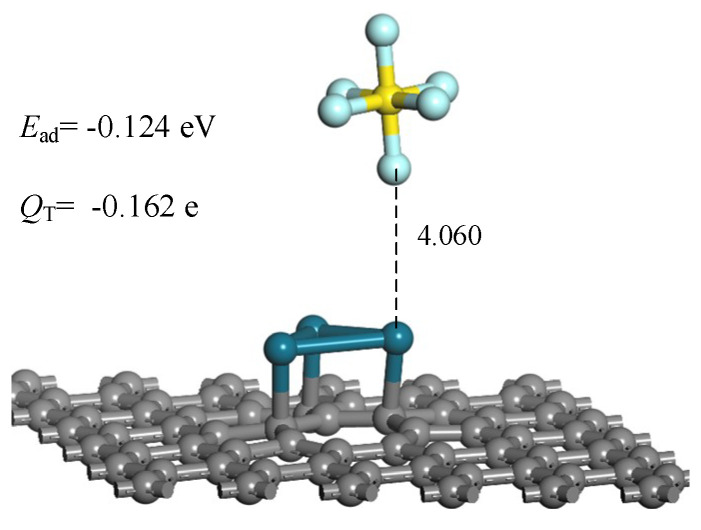
The most adsorption structure of SF_6_ on Pd_3_-graphene surface (distance in Å).

**Figure 11 sensors-20-04188-f011:**
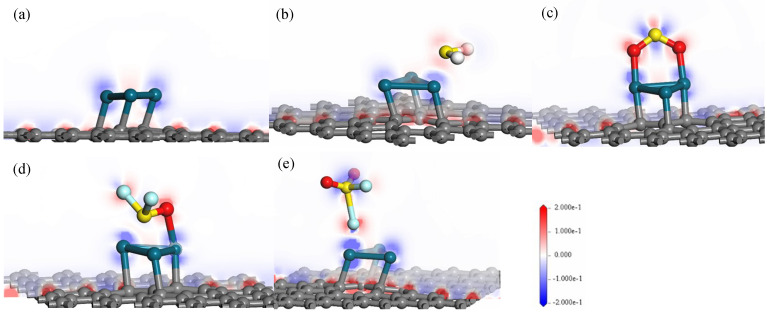
Deformation charge density (DCD) of pure Pd_3_-graphene and the most stable adsorption structures. (**a**) Pure Pd_3_-graphene; (**b**) H_2_S; (**c**) SO_2_; (**d**) SOF_2_; (**e**) SO_2_ F_2_.

**Figure 12 sensors-20-04188-f012:**
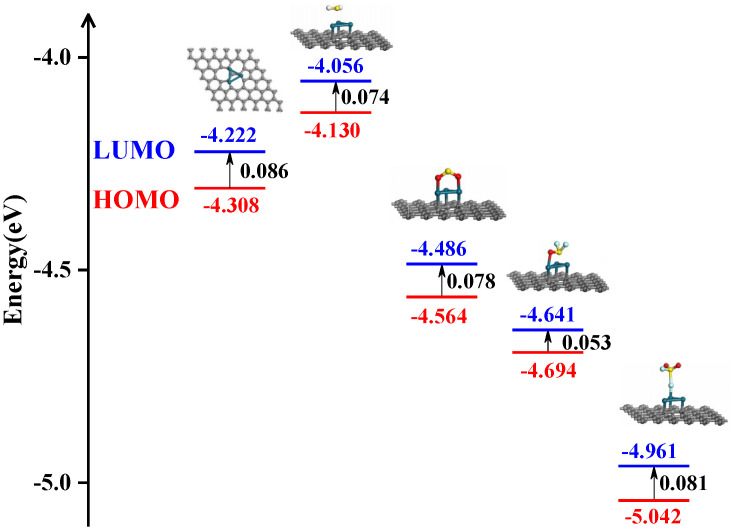
*E*_L_, *E*_H_ and *E*_g_ of pristine Pd_3_-graphene and various most stable adsorption systems. From the left to the right are pristine Pd_3_-graphene, M1, M4, M7 and M8 systems, respectively.

**Table 1 sensors-20-04188-t001:** Structural parameters of H_2_S adsorption systems on Pd_3_-graphene surface. Position represents the approaching way, for instance, S–Pd means the H_2_S molecule approaches the Pd atom by the S atom.

System	Position	*E*_ad_ (eV)	*Q*_T_ (e)	D (Å)
M1	S–Pd	−1.211	0.286	2.338
M2	H–Pd	−1.185	0.279	1.868

**Table 2 sensors-20-04188-t002:** Structural parameters of SO_2_ adsorption systems on Pd_3_-graphene surface. Position represents the approaching way, for instance, S–Pd means the SO_2_ molecule approaches the Pd atom by the S atom.

System	Position	*E*_ad_ (eV)	*Q*_T_ (e)	D (Å)
M3	S–Pd	−1.534	−0.023	2.192
M4	O–Pd	−1.591	−0.283	2.131

**Table 3 sensors-20-04188-t003:** Structural parameters of SOF_2_ adsorption systems on Pd_3_-graphene surface. Position represents the approaching way, for instance, S–Pd means the SOF_2_ molecule approaches the Pd atom by the S atom.

System	Position	*E*_ad_ (eV)	*Q*_T_ (e)	D (Å)
M5	S–Pd	−1.230	0.038	2.170
M6	F–Pd	−0.284	−0.255	2.124
M7	O–Pd	−1.282	−0.132	2.299

**Table 4 sensors-20-04188-t004:** Structural parameters of SO_2_ F_2_ adsorption systems on Pd_3_-graphene surface. Position represents the approaching way, for instance, F–Pd means the SO_2_ F_2_ molecule approaches the Pd atom by F atom.

System	Position	*E*_ad_ (eV)	*Q*_T_ (e)	D (Å)
M8	F–Pd	−0.920	−0.705	2.014
M9	O–Pd	−0.804	−0.294	2.181

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
