# Peer review of "Adsorption Properties of Pd3-Modified Double-Vacancy Defect Graphene toward SF6 Decomposition Products"

_sensors, 2020, doi:10.3390/s20154188_

Round 1
Reviewer 1 Report
This work reports the density functional theory (DFT) approach to determine the most stable adsorption systems on Pd3‐cluster‐doped 555‐777 graphene for use as a novel resistor‐type gas sensor to detect the SF6 decomposition products. This work is interesting in the computational perspective and the manuscript is well presented. This manuscript could be considered for publication after addressing the following comments. Therefore I recommend for major revision.
Reviewer comments
1. Does the size of Pd nanoclusters matter for gas adsorption? Do they have any influence on the gas adsorption (SF6) capacity, the conductivity, the energy gap (Eg) and the selectivity?
2. Is there any interference effect from other atmospheric gases? How to overcome the interference effect from other gases? What would be the effect of humidity and temperature on the adsorption capacity of SF6 decomposition products?
3. How to tune the Eg values and the sensitivity as well as the selectivity with respect to the graphene size and the nature of defects?
4. Any specific reason for considering Pd nanoclusters for this study? Why other noble metals like Au, Pt, Rh or Ru were not compared/discussed?
5. What would be the adsorption capacity of the defect free graphene? Any controls? Did you perform any calculations for 555‐777 graphene without doping Pd nanoclusters?
7. It is known that the graphene exists in the form of monolayer, bilayer and multillayers depending on the approach that is top-down or bottom-up used for their preparation. Would the adsorption capacity be then affected by the graphene layer thickness? Would the graphene size have any effect on the gas adsorption capacity?
8. Please note that the graphene sheets produced in the lab could be enriched with various defects such as sp2 defects (pentagonal and octagonal), point defects (zig-zag edge, armchair or pentagon rings, Stone–Wale defect, single vacancy) or line defects (grain boundary) and this could vary depending on the method of synthesis and the preparation environment (temperature or pressure) used. The authors should therefore discuss about (i) the role of intrinsic defects as well as the nature of defects introduced to the graphene surface, (ii) how these intrinsic defects will affect the gas adsorption capacity? and, (iii) how these defects need to be engineered to have the sensitivity enhancement for SF6 decomposition products?
9. The authors should compare the adsorption capacity as well as the sensitivity of SF6 products on Pd3‐cluster‐doped 555‐777 graphene with other metal doped defect-rich graphene reported so far.
Author Response
Dear reviewer 1:
Thank you so much for taking your time to review our manuscript and giving us the chance to revise the manuscript. We have taken your constructive suggestions to improve the quality of the manuscript. The changes in the revised manuscript have been marked by red color.
Sincerely yours,
Lei Pang
Address: School of Electrical Engineering, Xi’an Jiaotong University, Xi’an, Shaanxi Province 710049, China
Tel: +86-186-2938-9017
E-mail: panglei_2013@mail.xjtu.edu.cn
-------------------------------------------------------------------------------------------------------
The following is a point-to-point response to the comments:
Comment 1: Does the size of Pd nanoclusters matter for gas adsorption? Do they have any influence on the gas adsorption (SF6) capacity, the conductivity, the energy gap (Eg) and the selectivity?
Answer 1: The size of nanocluster can affect gas adsorption according to 10.1016/j.apsusc.2015.06.112 and 10.1016/j.apsusc.2018.07.219, in which single Pt atom and Pt3 nanocluster are doped on anatase (101) respectively. The results show that Pt3 nanocluster has a greater adsorption capacity on SF6 decomposition products. Thus, we decide to modify 555-777 graphene with the similar size of Pd nanoclusters. It is necessary to consider the influence of SF6 during the gas adsorption. And we have calculated the adsorption structure of SF6 on Pd3-graphene surface, the small adsorption energy (-0.124 eV) means that Pd3-graphene can not absorb SF6 gas stably, so SF6 will not influence the gas adsorption. Detailed analysis has been added to the article (line 236-248).
Comment 2: Is there any interference effect from other atmospheric gases? How to overcome the interference effect from other gases? What would be the effect of humidity and temperature on the adsorption capacity of SF6 decomposition products?
Answer 2: In practical applications, the gas-insulated switchgear (GIS) is filled with SF6 insulating gas. Although a trace amount of impurity gas will enter during the installation process, the content in the entire gas chamber is extremely low. In addition, there are certain adsorbents in the gas chamber to remove impurity gases. Thus, it is not necessary to consider the effect from other atmospheric gases. The moisture content in the GIS is extremely low, so the sensor is basically not affected by humidity during operation. Temperature is a significant factor affecting the sensitivity of gas sensor. In subsequent experiments, we will focus on the sensitivity of the gas sensor to decomposition products at different temperatures.
Comment 3: How to tune the Eg values and the sensitivity as well as the selectivity with respect to the graphene size and the nature of defects?
Answer 3: Different size and defects of graphene may cause various Eg. In our work, the graphene we designed is a 6×6 supercell. If we want to adjust Eg according to the size of graphene and the nature of defects, we need to design graphene with different defects and different sizes, and further find that the law of size and defects of graphene affecting the Eg. It is a challenging but meaningful task, thank you for your novel idea.
Comment 4: Any specific reason for considering Pd nanoclusters for this study? Why other noble metals like Au, Pt, Rh or Ru were not compared/discussed?
Answer 4: We have added the specific reason for considering Pd nanoclusters in our work. Some scholars have published studies on the adsorption properties of SF6 on the perfect graphene crystal surface modified by other metals (like Au, Pt, Rh). We think that it may not be innovative to use them to modify graphene again. (line 57-60)
Comment 5: What would be the adsorption capacity of the defect free graphene? Any controls? Did you perform any calculations for 555‐777 graphene without doping Pd nanoclusters?
Answer 5: Yes. Before we modified the 555-777 graphene with Pd3 cluster, we calculated the adsorption of SF6 products on intrinsic 555-777 graphene. The result is that the adsorption energies of H2S, SO2, SOF2 and SO2F2 are 0.912, 0.963, 0.625 and 0.473 eV respectively, indicating that the adsorption of SOF2 and SO2F2 belongs to physical adsorption. So intrinsic 555-777 graphene is not suitable to detect those products. Because the focus of our research is the adsorption performance of Pd3-graphene surface, the calculation of intrinsic graphene is not mentioned in this article.
Comment 7: It is known that the graphene exists in the form of monolayer, bilayer and multillayers depending on the approach that is top-down or bottom-up used for their preparation. Would the adsorption capacity be then affected by the graphene layer thickness? Would the graphene size have any effect on the gas adsorption capacity?
Answer 7: According to 10.1016/j.apsusc.2017.01.286, the resistance of graphene will decrease significantly as the layers increases, but this is not good for improving the sensitivity and selectivity of Pd3-graphene gas sensor. The lower the resistance, the higher the sensitivity of the electrical signal we need to detect, which will put forward better requirements for our detecting equipment. Certainly, it is not easy to prepare high-purity single-layer graphene bins, but we will strictly control the number of graphene layers to ensure the accuracy and reliability of the experiment. We calculated the adsorption positions the graphene apart from the doping position, finding that the adsorption energy did not change compared with the intrinsic 555-777 graphene. So it is believed that the performance of the sensor mainly depends on the modified positions, changing the size of other regions has little effect on improving its adsorption properties.
Comment 8: Please note that the graphene sheets produced in the lab could be enriched with various defects such as sp2 defects (pentagonal and octagonal), point defects (zig-zag edge, armchair or pentagon rings, Stone–Wale defect, single vacancy) or line defects (grain boundary) and this could vary depending on the method of synthesis and the preparation environment (temperature or pressure) used. The authors should therefore discuss about (i) the role of intrinsic defects as well as the nature of defects introduced to the graphene surface, (ii) how these intrinsic defects will affect the gas adsorption capacity? (iii) how these defects need to be engineered to have the sensitivity enhancement for SF6 decomposition products?
Answer 8: In the paper, we focus on the effect of Pd3 cluster during gas adsorption and ignore the role of the intrinsic defect and its influence. Thanks for your suggestion, we have analyzed the nature of the defect and its impact during the adsorption process in Electronic properties part (line 250-259). In our work, we can conclude that graphene with double-vacancy defect is a potential basic material employing in the detection of SF6 products. If we want to explore how to engineer various defects to enhance the sensitivity of gas sensor, we need to design different defects on perfect graphene and compare the difference of them just like the design in Answer 3. It can be a meaningful and interesting research.
Comment 9: The authors should compare the adsorption capacity as well as the sensitivity of SF6 products on Pd3‐cluster‐doped 555‐777 graphene with other metals doped defect-rich graphene reported so far.
Answer 9: We have compared our results with the previous research reported in 10.1109/ACCESS.2019.2945469, and concluded that Pd3-graphene can be better sensor material on detecting SF6 products (line 298-304). We really appreciate for your review.
Reviewer 2 Report
In this work, the authors have explored the potential of Pd3-graphene as a novel gas sensor of SF6 decomposition components based on DFT calculations. The result suggests that Pd3 cluster doped 555-777graphene could be employed as a new gas sensor. The idea of this study is interesting and worth for publication in Sensors. Moreover, this paper lays a foundation to expand applications of graphene gas sensors and put forward a new method to detect SF6 decomposition components. Thus, I recommend publishing it after minor modifications:
Comment 1: Some values in Figures are too small to observe. They should be written with larger size.
Comment 2: There are some inappropriate wordings in the manuscript.
1) In Figure 1, the “Pd3-graphene” in the figure caption should be “Pd3-graphene”. Please take care of the subscript.
2) “While H2S and SO2 absorbs on the Pd3-graphene surface in the most stable configuration”, “absorbs” should be “absorbing”.
Comment 3: In DFT calculations, smearing is an important parameter to ensure the accuracy of the results. Please add the specific value of it in the Computational Details and Models part.
Comment 4: In the introduction part, “Graphene is a 2D material with a unique 2D monoatomic layer structure and an electronic energy band structure. Its excellent characteristics, including high electron mobility, high thermal conductivity, brilliant mechanical properties, and large specific surface area, make it a promising gas sensor material” , the author should mention the latest literature, such as: Micromachines 2020, 11(1), 58; Physica E: Low-dimensional Systems and Nanostructures 117 (2020) 113840.
Author Response
Dear reviewer 2:
Thank you so much for taking your time to review our manuscript and giving us the chance to revise the manuscript. We have taken your constructive suggestions to improve the quality of the manuscript. The changes in the revised manuscript have been marked by red color.
Sincerely yours,
Lei Pang
Address: School of Electrical Engineering, Xi’an Jiaotong University, Xi’an, Shaanxi Province 710049, China
Tel: +86-186-2938-9017
E-mail: panglei_2013@mail.xjtu.edu.cn
-------------------------------------------------------------------------------------------------------
The following is a point-to-point response to the comments:
Comment 1: Some values in Figures are too small to observe. They should be written with larger size.
Answer 1: We have rewritten the numbers in Figures to make sure that its size is suitable.
Comment 2: There are some inappropriate wordings in the manuscript.
1) In Figure 1, the “Pd3-graphene” in the figure caption should be “Pd3-graphene”. Please take care of the subscript.
2) “While H2S and SO2 absorbs on the Pd3-graphene surface in the most stable configuration”, “absorbs” should be “absorbing”.
Answer 2: We have done our best to revise the whole manuscript, including the errors you mentioned. (line 99)
Comment 3: In DFT calculations, smearing is an important parameter to ensure the accuracy of the results. Please add the specific value of it in the Computational Details and Models part.
Answer 3: The smearing we utilized in the DFT calculations is 0.001Ha, and we have added it into the Computational Details and Models part. (line 73-74)
Comment 4: In the introduction part, “Graphene is a 2D material with a unique 2D monoatomic layer structure and an electronic energy band structure. Its excellent characteristics, including high electron mobility, high thermal conductivity, brilliant mechanical properties, and large specific surface area, make it a promising gas sensor material” , the author should mention the latest literature, such as: Micromachines 2020, 11(1), 58; Physica E: Low-dimensional Systems and Nanostructures 117 (2020) 113840.
Answer 4: We read the article you recommended and found that it is the latest research about graphene sensors, so we cited it in our article. We really appreciate for your review.
Round 2
Reviewer 1 Report
The authors have addressed the reviewers comments and the quality of the manuscript has been improved. Therefore, I recommend the revised version of the manuscript for publication in Sensors.
Author Response
Dear reviewer :
Thank you so much for taking your time to review our manuscript.
Sincerely yours,
Lei Pang
Address: School of Electrical Engineering, Xi’an Jiaotong University, Xi’an, Shaanxi Province 710049, China
Tel: +86-186-2938-9017
E-mail: panglei_2013@mail.xjtu.edu.cn